# Gut microbial bile and amino acid metabolism associate with peanut oral immunotherapy failure

Mustafa Özçam [1], Din L. Lin[1], Chhedi L. Gupta[1,2,9], Allison Li[1], J. Carlos Gomez [1], Lisa M. Wheatley[3], Carolyn H. Baloh[4,5,6], Srinath Sanda[7], Stacie M. Jones[8] & Susan V. Lynch [1] ✉

Peanut Oral Immunotherapy (POIT) holds promise for remission of peanut allergy, though treatment is protracted and successful in only a subset of patients. Because the gut microbiome has been linked to food allergy, we sought to identify fecal predictors of POIT efficacy and mechanistic insights into treatment response. Here, we conducted a secondary analysis of the IMPACT randomized, double-blind, placebo-controlled POIT trial (NCT01867671), using longitudinal fecal samples from 90 children, and performed 16S rRNA sequencing, shotgun metagenomics, and untargeted metabolomics. Integrated multi-omics analyses revealed a relationship between gut microbiome metabolic capacity and treatment outcomes. Five fecal bile acids present prior to treatment initiation predicted POIT efficacy (AUC 0.71). Treatment failure was associated with a specific bile acid profile, enhanced amino acid utilization, and higher copy number of the *ptpA* gene encoding a bacterial hydrolase that cleaves tripeptides containing proline residues – a feature of immunogenic peanut Ara h 2 proteins. In vitro, peanut-supplemented fecal cultures of children for whom POIT failed to induce remission evidenced reduced Ara h 2 concentrations. Thus, distal gut microbiome metabolism appears to contribute to POIT failure.

Peanut protein allergy (PA), triggered by allergenic Ara h proteins[1], is the leading cause of food-induced anaphylaxis[2]. Until the 2020 U.S. Food and Drug Administration (FDA) approval of peanut oral immunotherapy (Palforzia™) for patients over 4 years old, strict avoidance of peanuts and peanut-containing products has been the primary management strategy, with the exception of a few limited facilities conducting OIT as clinical research. Peanut oral immunotherapy (POIT) has emerged as a widely used treatment for PA[3]. Involving gradual oral introduction of increasing concentrations of peanut powder, POIT induces desensitization, defined as an increase in reaction threshold while on treatment, in ~50–70% of treated patients. While POIT has demonstrated efficacy in desensitizing patients to peanuts, the induction of remission, defined as the prolonged absence of clinical reactivity after treatment cessation, is observed in a smaller subset of

[1]Division of Gastroenterology, Department of Medicine, University of California, San Francisco, San Francisco, CA, USA. [2]Benioff Center for Microbiome Medicine, Department of Medicine, University of California, San Francisco, San Francisco, CA, USA. [3]National Institute of Allergy and Infectious Diseases, National Institutes of Health, Bethesda, MD, USA. [4]The Immune Tolerance Network, Boston, MA, USA. [5]Division of Allergy and Clinical Immunology, Brigham and Women's Hospital, Boston, MA, USA. [6]Harvard Medical School, Boston, MA, USA. [7]The Immune Tolerance Network, San Francisco, CA, USA. [8]Division of Allergy and Immunology, Department of Pediatrics, University of Arkansas for Medical Sciences and Arkansas Children's Hospital, Little Rock, AR, USA. [9]Present address: ICMR-National Institute of Immunohaematology, Chandrapur Unit (ICMR-CRMCH), Chandrapur, Maharashtra, India. ✉e-mail: susan.lynch@ucsf.edu

~20–30% of POIT-treated patients[4,5]. POIT cost, prolonged duration of treatment (several years), and burden of daily dosing highlight the need for improved predictive markers of outcome and adjunctive therapies to increase rates of remission.

The IMPACT, Oral Immunotherapy for the Induction of Tolerance and Desensitization in Peanut-Allergic Children trial (NCT01867671) was the first randomized, double-blinded, placebo-controlled, multi-center clinical trial to evaluate the efficacy and safety of POIT in peanut-allergic children ages 12–48 months old. While 84% of the children receiving POIT achieved desensitization, only 29% achieved remission following POIT discontinuation and 26 weeks of peanut avoidance. The IMPACT trial yielded three clinical outcomes among peanut-allergic children treated with POIT: (i) those who achieved both desensitization and remission (D+R+), (ii) those who achieved desensitization but not remission (D+R−), (iii) those who did neither achieved desensitization nor remission (D−R−). Notably, younger age and lower peanut-specific serum IgE concentrations at the outset of the trial were more likely to result in a D+R+ outcome[6]. However, the factors driving divergent POIT outcomes remain poorly understood.

Emerging evidence underscores the critical role of the gut microbiome in shaping immune responses and influencing the development of allergic diseases, including food allergy[7,8]. Perturbations to infant gut microbiome composition and functional capacity have been linked to increased susceptibility to allergic sensitization and impaired immune tolerance to food allergens in later childhood[9,10]. Infant gut microbiomes of those who subsequently develop allergic disease exhibit distinct metabolic profiles that can induce allergic inflammation in vitro[11]. Additionally, relationships between fecal metabolic profiles and food allergy have been described in older children[12–14], indicating that fecal metabolic dysfunction is a consistent characteristic of disease development and incidence. Gut microbiomes modulate host immunity through the production of metabolites[15,16], including those that affect immunotherapy efficacy[17,18]. Specific microbial metabolite classes, such as bile acids, play an essential role in regulating immune homeostasis and promoting regulatory pathways necessary for allergen tolerance[16,19,20]. Moreover, both age[21] and allergic sensitization status[11] are closely related to early-life gut microbiome composition and metabolic activity[22]. Thus, we hypothesized that gut microbiome features prior to initiation of POIT are associated with treatment outcomes and that the longitudinal assessment of fecal microbiomes from children in this trial would reveal mechanisms underlying variance in POIT efficacy. In this work, we show that increased gut microbial protein hydrolysis capacity and peanut protein degradation in parallel with a decreased abundance of fecal amino acid metabolites and a distinct bile acid profile are associated with POIT-failure.

## Results

### Study population and clinical trial outcome

Details of the IMPACT POIT clinical trial design have been previously published[6]. Briefly, at baseline, 146 peanut-allergic children were enrolled and randomized (2:1) to either POIT or placebo treatment. After a dose escalation phase of 30 weeks, children in the POIT arm received 2000 mg peanut powder (lightly roasted, partly defatted [12% fat]) while the placebo group received oat flour for 104 weeks (total blinded treatment period 134 weeks). Participants who passed the 5 g peanut powder, double-blind, placebo-controlled, food challenge (DBPCFC) at the end of treatment (week 134) were categorized as desensitized (D+). Independent of the DBPCFC outcome at week 134, all participants avoided peanut consumption for 26 weeks (avoidance period), and those who passed the 5 g peanut powder DBPCFC at the end of this avoidance period (week 160) were categorized as being in remission (R+).

Based on DBPCFC results at the end of treatment and end of avoidance, the IMPACT clinical trial yielded three outcome groups:

Desensitized and Remission (D+R+), Desensitized and No Remission (D+R−), or Not Desensitized and No Remission (D−R−; Fig. 1a). Ninety-three out of 146 participants adhered to the study protocol (per protocol group) until the end of the avoidance. Of these, 90 provided longitudinal fecal samples at five time points: baseline (prior to POIT initiation), end of buildup (EoB), mid-maintenance (MM), end of treatment (EoT), and the end of avoidance (EoA) (Supplementary Fig. 1a and Supplementary Data 1) resulting in a total of 327 samples included in this study. Participant baseline characteristics including age, sex, study locations, antibiotic usage history, and atopic comorbidities are reported in Supplementary Data 2[6].

### Fecal microbiota composition associates with peanut oral immunotherapy outcomes

Participants who completed the IMPACT trial ($n = 90$ participants) did not differ in age between POIT and placebo-treated groups (Supplementary Fig. 1b). Consistent with observations made in the parent clinical trial[6], within the POIT-treated group, D+R+ participants were significantly younger compared to those within the two other outcome groups (D+R− and D−R−; Supplementary Fig. 1c). Two hundred sixty-three fecal samples (Placebo, $n = 73$; POIT, $n = 190$) from 90 participants yielded high-quality 16S rRNA amplicon sequence data (see the "Methods" section, Supplementary Fig. 1a and Supplementary Data 1). Comparing fecal bacterial phylogenetic diversity (α-diversity) (Fig. 1b and Supplementary Fig. 1d) and composition (β-diversity) (Fig. 1e and Supplementary Data 3), over time revealed no significant difference between the placebo and POIT participants at any time point indicating that POIT does not appreciably alter fecal microbiota composition. Similarly, fecal microbiota composition was not different between placebo and individual POIT-outcome groups in pairwise comparisons (Supplementary Data 3). Therefore, for the remainder of the study, we focused on the POIT arm of the trial to identify predictors and mechanisms of treatment outcomes.

To determine potential confounding factors within the POIT arm, clinical and demographic variables were examined as independent terms at each time point using two-sided PERMANOVA based on an unweighted UniFrac distance matrix. Age at screening, sample collection date, sex, and study site location significantly related to variance in fecal microbiota composition at various time points throughout the trial (Supplementary Data 4). Thus, subsequent statistical analyses were adjusted for these covariates. Both POIT outcome (3 groups; $P = 0.008$, $R^2 = 0.07$, $n = 47$) and peanut allergy remission (R+ status; $P = 0.003$, $R^2 = 0.04$, $n = 47$) also associated with variance in fecal microbiota composition in baseline samples (Supplementary Data 4), indicating that pre-treatment microbiota related to POIT efficacy.

### Children who develop POIT-induced remission exhibit a distinct fecal microbiota composition throughout the course of the trial

Although longitudinal α-diversity was not different between the three POIT-outcome groups within the POIT arm (Linear mixed-effect model [LME]; Fig. 1c), the D+R+ group exhibited significantly lower phylogenetic diversity at baseline compared to D+R− and D-R- groups ($P = 0.001$ and $P = 0.052$, respectively; Wilcoxon rank-sum test. Figure 1d and Supplementary Fig. 1d). This finding remained significant despite adjustment for age (Fig. 1c, $P = 0.043$ ANOVA). Additionally, POIT outcome groups exhibited differences in fecal microbiota β-diversity (Fig. 1f). Specifically, participants who achieved remission (D+R+) exhibited differences in fecal microbiota composition along the first principal component (axis 1) compared to those who did not (D+R+ vs. D+R−, $P = 0.01$; D+R+ vs. D−R−, $P = 0.004$, LME; Fig. 1g) throughout the course of the trial. This provided evidence that both fecal microbiota composition and diversity are associated with POIT outcomes.

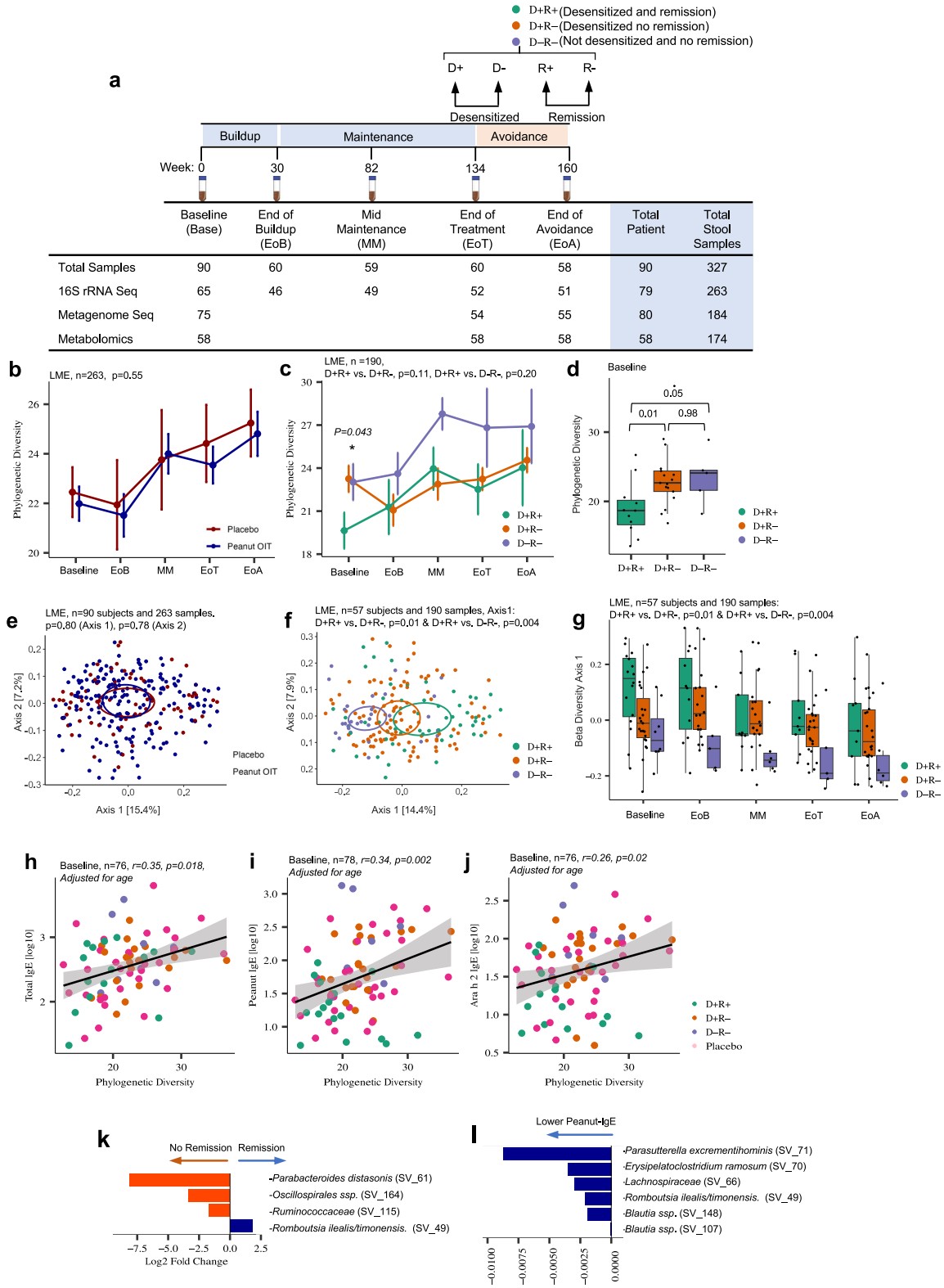

## Baseline bacterial phylogenetic diversity positively correlates with peanut-specific IgE irrespective of age

In the IMPACT clinical trial, lower concentrations of peanut-specific serum Immunoglobulin E (IgE) prior to POIT initiation were predictive of remission[6]. Thus, we sought to determine whether fecal microbiota features related to IgE levels. In age-adjusted analyses, positive correlations between α-diversity and serum levels of total IgE (Fig. 1h),

peanut-specific IgE (Fig. 1i) and Ara h 2-specific IgE (Fig. 1j; $P < 0.05$ Pearson correlation) were observed in baseline samples, suggesting that increased fecal bacterial diversity relates to higher IgE levels associated and reduced likelihood of remission following POIT in peanut-allergic children.

Since both younger age and lower baseline peanut-specific IgE levels predicted clinical remission in the IMPACT trial, we next

**Fig. 1 | Fecal microbiota composition and diversity associated with peanut oral immunotherapy outcomes and peanut allergy severity. a** Schematic overview of IMPACT trial fecal microbiome study[9]. Numbers represent the number of fecal samples in each dataset and time point. **b** No significant difference in α-diversity (Faith's phylogenetic diversity) was observed over the course of the IMPACT trial between Peanut Oral Immunotherapy (POIT) and placebo arms ($\beta = -0.84$, SE = 1.40, df = 166, $t = -0.60$, $P = 0.55$, two-sided linear mixed-effects model). Data are presented as mean values ± SEM. **c** No significant difference in α-diversity (Faith's phylogenetic diversity) was observed over the course of the IMPACT trial between the POIT outcome groups (D+R−, D−R−, and D+R+), after adjustment for age using a two-sided linear mixed-effects model with random intercepts for subject (D+R− vs. D+R+: $\beta = 2.58$, $t = 1.59$, $P = 0.115$ and D−R− vs. D+R+: $\beta = 2.69$, $t = 1.26$, $P = 0.210$). **d** However, at baseline, prior to POIT initiation, the D+R+ group exhibited significantly lower phylogenetic diversity compared to either the D+R− and D−R− groups. Wilcoxon signed-rank test (two-sided, $n = 47$; D+R+ = 16, D+R− = 23, D−R− = 8). Data are presented as mean values ± SEM. Boxplots show the median (center line), 25th and 75th percentiles (box bounds), and whiskers extend to values within 1.5× the interquartile range. **e** Although, the longitudinal fecal microbiota composition (unweighted Unifrac distance matrix) was similar between POIT and placebo arms (two-sided linear mixed-effect model, $n = 90$ subjects and 263 samples. $P = 0.80$ for Axis 1, and $P = 0.78$ for Axis 2). **f** Significantly different fecal microbiota composition was observed within the POIT outcome groups (two-sided linear mixed-effect model, $n = 57$ subjects and 190 samples, Axis 1: D+R+ vs. D+R−, $p = 0.01$ and D+R+ vs. D−R−, $p = 0.004$). **g** D+R+ group exhibited significantly different fecal microbiota composition compared to both D+R− ($p = 0.01$) and D−R − ($p = 0.004$) groups throughout the trial (two-sided linear mixed-effect model). Data are presented as mean values ± SEM. Boxplots show the median (center line), 25th and 75th percentiles (box bounds), and whiskers extend to values within 1.5× the interquartile range. **h** Baseline gut bacterial phylogenetic diversity positively correlates with baseline total IgE, **i** peanut-specific IgE, and **j** Ara h 2-specific IgE levels, respectively. Participants who enrolled in the IMPACT trial but did not complete the trial were included as 16S rRNA sequencing and clinical serum IgE levels were available (two-sided Pearson correlation, $P < 0.05$, adjusted for age). **k** Baseline differentially abundant bacterial taxa between children who achieved remission ($n = 16$) versus no remission ($n = 31$). Two-sided linear mixed-effect model ($P.FDR < 0.05$, adjusted for age). **l** Baseline Peanut-IgE associated bacterial taxa ($n = 47$). Two-sided linear mixed-effect model ($P.FDR < 0.05$). Estimate represents the predicted value of the effect size or relationship derived from the statistical model, reflecting the magnitude and direction of the association. EoB: End of Buildup, MM: Mid Maintenance, EoT: End of Treatment, and EoA: End of Avoidance. D+R+: Desensitized and Remission, D+R−: Desensitized no Remission, D−R−: Not desensitized and no Remission. LME: Linear mixed-effect model. Source data are provided as a Source Data file.

identified baseline Sequence Variances (SVs) associated with both peanut-specific IgE level and remission status in age-adjusted analyses. *Romboutsia ilealis/timonensis* was associated with POIT-induced remission, while *Ruminococcaceae* along with *Parabacteroides distasonis*, and *Oscillospirales* members associated with failure to develop remission ($P.FDR < 0.05$. *LME*, adjusted for age, Fig. 1k). *R. ilealis/ timonensis* was also negatively associated with peanut-specific IgE levels at baseline ($P.FDR < 0.05$. Two-sided LME, adjusted for age); Fig. 1l and Supplementary Data 5) and with peanut- and component (Ara h)-specific IgE levels (Ara h 1, 2, 3 and 6 IgE; Supplementary Fig. 1e). Thus, though POIT does not significantly impact fecal microbiota composition, the diversity and composition of the microbiota prior to treatment initiation and throughout the IMPACT trial relate to treatment outcomes. Moreover, lower bacterial phylogenetic diversity and the relative abundance of specific fecal microbial members prior to treatment, associated with multiple measures of peanut allergic sensitization, irrespective of participant age, in peanut-allergic children.

## Baseline bile acid profile associates with POIT efficacy

Fecal microbiota perturbation and metabolic dysfunction, including increased concentrations of metabolites that promote cardinal features of allergic inflammation, are characteristic of allergic disease[11,23–25]. To determine whether the distinct fecal microbiota compositions associated with POIT outcomes exhibited divergent metabolic profiles, untargeted metabolomic analyses were performed on a subset (see the "Methods" section) of participants who provided fecal samples with sufficient remaining material for analysis at all three key visits: baseline, end of treatment, and end of avoidance ($n = 58$ participants [POIT = 43, Placebo = 15], 174 fecal samples; Fig. 1a, Supplementary Fig. 1a and Supplementary Data 1). Like fecal microbiota composition, baseline fecal metabolite profiles were associated with POIT outcome groups ($n = 43$, $R^2 = 0.07$, $P = 0.01$), and, more specifically, with remission status within POIT-treated children ($n = 43$, $R^2 = 0.04$, $P = 0.01$; two-sided PERMANOVA, Euclidean distance matrix, Supplementary Data 6).

To identify metabolites that relate to POIT outcomes, a data reduction approach, weighted gene correlation network analyses (WGCNA), was applied to identify modules of co-associated metabolites that were then related to POIT-outcomes. Fifty metabolite modules (untargeted metabolite modules [UMMs]; Supplementary Data 7) were identified, nine of which were significantly associated with POIT

outcomes ($P.FDR < 0.05$, ANOVA, adjusted for age, Fig. 2a and Supplementary Data 8). These POIT outcome-associated metabolic modules mostly comprised of lipids, specifically bile acids (BA), and amino acid modules (AA; Fig. 2a and Supplementary Data 7). Two-sided PERMANOVA analyses showed that BA profiles significantly differed between POIT-outcome groups only at baseline (Fig. 2b; $n = 43$, $R^2 = 0.10$, $P = 0.015$, Two-sided, PERMANOVA, Euclidean distance matrix), but not at the end of treatment or avoidance (Supplementary Fig. 2b), indicating that pre-treatment fecal metabolic status appears most related to treatment outcome. Three BA modules (UMM10, UMM15, and UMM4) are associated with treatment outcomes (Fig. 2c–e). UMM10, comprised of lithocholate and deoxycholate amongst other BAs, was increased in children who did not develop POIT-induced remission (Fig.2c, f). UMM15 containing sulfated-BAs (Fig. 2d, g), and UMM4, comprised of 7-ketolithocholate and 7-ketodeoxycholate amongst other BAs were decreased in these participants (Fig. 2e, h). Together, these data suggest that the specific BA profile present at the initiation of POIT associated with POIT efficacy.

Since BAs are drivers of gut microbiota maturation in early life[26], we next investigated whether relationships existed between the UMM10 and UMM15 BA modules and fecal microbiota features that are associated with POIT-outcomes. UMM15, primarily comprised of bacterial-derived secondary BAs (Supplementary Data 7), exhibited a significant negative relationship with fecal microbiota α-diversity (lower baseline α-diversity is associated with remission; Fig. 1d) and a positive correlation with axis 1 of the baseline microbiota composition. In contrast, the UMM10 module exhibited the opposite relationship, being positively correlated with fecal α-diversity and negatively correlated with axis 1 of the baseline microbiota composition (Supplementary Fig. 2c). These data suggest that at baseline, secondary BAs associated with the lower fecal bacterial diversity and a distinct fecal bacterial composition that characterize children who develop POIT-induced remission.

Although 16S rRNA-based biomarker sequencing allows relationships between fecal microbiota and clinical outcomes to be uncovered, because it is based on a single gene, it fails to provide information on microbiome gene content and functional contribution to treatment outcome[27]. To identify the fecal microbial pathways associated with POIT outcomes including those responsible for the metabolic differentials observed across treatment outcome groups, shotgun metagenomic sequencing data was generated on baseline ($n = 75$), end of treatment, ($n = 54$), and end of avoidance ($n = 55$) fecal samples including

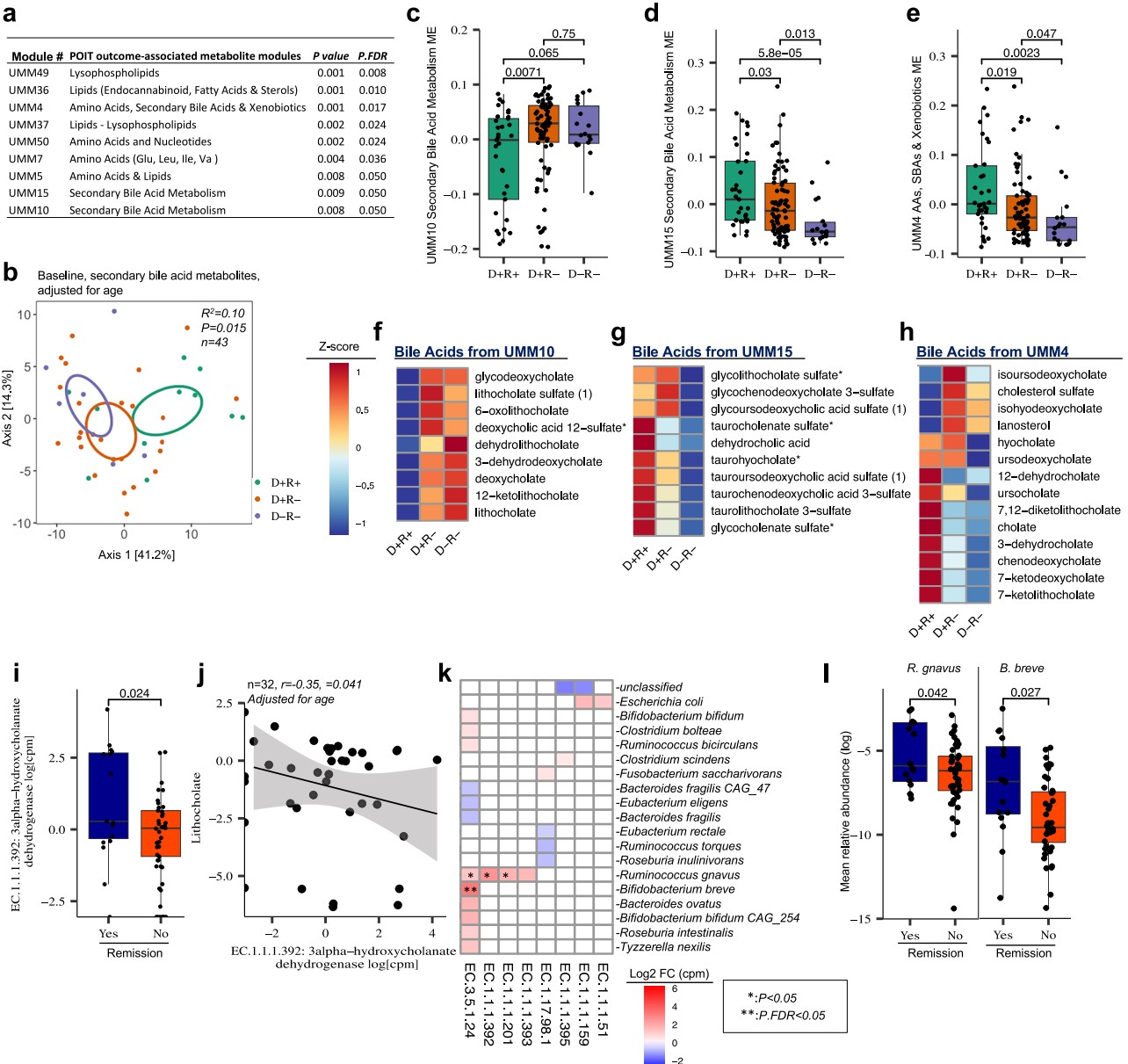

**Fig. 2 | Baseline bile acid profile associated with POIT efficacy. a** Association between untargeted metabolomics module (UMM) eigenvectors and POIT outcomes (two-sided *ANOVA*, adjusted for age). **b** Ordination of baseline secondary BA metabolites ($n = 43$, $R^2 = 0.10$; $P < 0.015$, adjusted for age), PERMANOVA analyses (two-sided) based on Euclidean distance matrix. **c** Difference in Module Eigenvectors (ME), which were determined based on WGCNA analyses (see the "Methods" section) and represents a measure of the joint abundance profile of a specific module, of UMM10. **d** UMM15, and **e** UMM4 between POIT-outcome groups. Two-sided Wilcoxon signed-rank test ($n = 129$; D+R+ = 33, D+R− = 78, D−R− = 18). Data are presented as mean values ± SEM. Boxplots show the median (center line), 25th and 75th percentiles (box bounds), and whiskers extend to values within 1.5× the interquartile range. **f** Baseline *Z*-scores of each BA-related metabolite of UMM10. **g** UMM15 and **h** UMM4 in each POIT-outcomes ($n = 43$; D+R+ = 11, D+R− = 26, D−R− = 6). Blue colors represent low *z*-scores thus low abundance and red colors represent high *z*-score and higher abundance. **i** POIT responders exhibit an increased copy number of the EC.1.1.1.392 enzyme compared to non-responders at baseline ($P = 0.024$, Two-sided Wilcoxon signed-rank test; Remission, Yes = 16, No = 44). Data are presented as mean values ± SEM. Boxplots show the median (center line), 25th and 75th percentiles (box bounds), and whiskers extend to values

within 1.5× the interquartile range. **j** Copy number of EC.1.1.1.392 enzyme negatively correlates with lithocholate abundance at baseline samples. Two-sided Pearson correlation ($n = 32$, $r = −0.35$, $p = 0.041$. Adjusted for age). The shaded area around the line represents the 95% confidence interval for the fitted regression line. **k** *Bifidobacterium breve* and *Ruminococcus gnavus* encode significantly higher copy numbers of secondary BA production enzymes in POIT responders ($n = 16$) compared to non-responders ($n = 44$). Data were filtered to retain enzymes with log2-fold changes exceeding ±0.5. Asterisk "*" represents $P < 0.05$, and double Asterisk "**" represents $P.FDR < 0.05$. Log2 FC (cpm) represents the Log2 fold change of copies per million between remission and no-remission groups (two-sided linear mixed effect models). Red color represents an increased Log2 FC (cpm) in the remission group. Exact $P$ values are presented in Supplementary Data 12. **l** The mean relative abundances of *R. gnavus* ($P = 0.042$) and *B. breve* ($P = 0.027$) are significantly enriched in POIT responders ($n = 16$) compared to non-responders ($n = 44$). Data are presented as mean values ± SEM. Boxplots show the median (center line), 25th and 75th percentiles (box bounds), and whiskers extend to values within 1.5× the interquartile range. Statistical comparisons were performed using the two-sided Wilcoxon rank-sum test. Source data are provided as a Source Data file.

all samples that had undergone parallel untargeted metabolomic analysis (Fig. 1a, Supplementary Fig. 1a, and Supplementary Data 1). Like 16S rRNA-based fecal microbiota and metabolome composition, within the POIT arm, fecal microbiome functional capacity at baseline was associated with POIT outcome groups ($n = 60$, $R^2 = 0.04$, $P = 0.025$), and with remission status ($n = 60$, $R^2 = 0.022$, $P = 0.0027$; PERMANOVA, Canberra distance matrix, Supplementary Data 9).

Next, we examined fecal metagenomes at the E.C. level, particularly focusing on those enzymes known to play a role in gut microbial BA metabolism, including the well-described 7α-dehydroxylation enzymes encoded by the bacterial *bai* operon[28,29] (Supplementary Data 10). Amongst these, only one gut microbial encoded BA enzyme, EC. 1.1.1.392 (3-alpha-hydroxycholanate dehydrogenase) which utilizes lithocholate as a substrate to produce iso-BAs[30], significantly differed (*P.FDR* < 0.05, log2 fold change ≥ |0.5|, LME. Supplementary Data 11) at baseline, being increased in POIT-responders (Fig. 2i). Consistent with enhanced microbial utilization of lithocholate, the abundance of the bacterial gene encoding this enzyme negatively correlated with the relative concentration of lithocholate (Fig. 2j) in baseline fecal samples.

Using metagenomic data, we next identified microbial species encoding BA enzymes, including EC.1.1.1.392. Notably, *Bifidobacterium breve* and *Ruminococcus gnavus* exhibited higher gene copy numbers for enzymes involved in BA metabolism. These included choloylglycine hydrolase (EC. 3.5.1.24, *P.FDR* < 0.05) and enzymes linked to iso-BA production (EC. 1.1.1.392, EC. 1.1.1.393) as well as 7-β-hydroxysteroid dehydrogenase (EC. 1.1.1.201, *P* < 0.05 but *P.FDR* > 0.05, LME) in patients who achieved POIT-induced remission (Fig. 2k and Supplementary Data 12). Taxonomic analysis using Kraken2 further revealed that both *B. breve* and *R. gnavus* were significantly enriched in POIT-responsive children (Fig. 2l). These findings suggest that the distinct fecal BA composition between POIT responders and non-responders is driven by enhanced microbial BA metabolism capacity in POIT-responsive patients.

### Abundance of select pre-treatment bile acids predicts peanut oral immunotherapy outcomes

We next performed an integrative data analysis on metagenomic and paired metabolomic datasets using Multi-Omics Factor Analyses (MOFA2). This analysis identified seven distinct factors (Supplementary Fig. 3a), five of which significantly differentiated POIT response groups (ANOVA, $P < 0.05$; Supplementary Fig. 3b). Several of these factors, e.g. Factor 3, included microbial pathways for AA biosynthesis that were enriched in those who achieved POIT-induced remission. In contrast, Factor 2, which included gluconeogenesis and anaerobic energy metabolism among the top five microbial pathways contributing to factor weight, was the most significant differential factor between all POIT outcome groups (D+R+ vs. D+R−: $P = 0.0028$, D+R+ vs. D−R−: $P = 0.00045$ and D+R− vs. D−R−: $P = 0.043$ Wilcoxon signed-rank test, Fig. 3a, b; Supplementary Fig. 3c) and significantly enriched in the fecal microbiome of children who did not develop POIT-induced remission (Supplementary Fig. 3b). In addition, primary and secondary BA metabolites including 7-ketodeoxycholate and 7-ketolithocholate were amongst the top 5 metabolites in Factor 2, all of which were depleted in those who did not achieve peanut allergy remission (Fig. 3c).

To determine whether the top five metabolites contributing to the weight of Factor 2, including 7-ketodeoxycholate and 7-ketolithocholate could serve as predictive markers for POIT-induced remission, a machine learning approach using a logistic regression model was applied. The baseline fecal abundance of these five metabolites produced a moderate predictive ability (area under the curve (AUC) from 100 times repeated five-fold cross-validation, measured as mean AUC ± standard deviation (s.d.: $AUC_{logistic\ regression}$: 0.712 ± 0.081; Fig. 3d). To confirm our findings, a second machine learning model employing a random forest model was applied and demonstrated similar performance (Supplementary Fig. 3e and Supplementary Data 13).

Among those predictive BAs, 7-ketodeoxycholate and 7-ketolithocholate are produced from deoxycholate and lithocholate[31] respectively, both of which were depleted in POIT-responders. The NADP+-dependent gut microbial EC. 1.1.1.201: 7-beta-hydroxysteroid dehydrogenase enzyme (7b-HSDH) plays a critical role in this pathway[32]. The copy number of the gene encoding EC. 1.1.1.201 was also significantly enriched in fecal microbiomes of those who achieved remission ($P < 0.05$, Fig. 2k), and positively correlated with the abundance of 7-ketodeoxycholate and 7-ketolithocholate secondary BAs (Fig. 3f, g, $P < 0.05$; Pearson correlations). Overall, our data indicates that fecal concentrations of select BAs including 7-ketodeoxycholate and 7-ketolithocholate prior to initiation of POIT, represent a useful predictor of treatment response and identify the specific gut bacteria and enzymes responsible for their production.

### Enhanced microbiome protein metabolism is associated with POIT failure

Four out of nine of POIT-associated metabolite modules (UMM4, UMM5, UMM7, and UMM50) were primarily comprised of AAs (Fig. 2a and Supplementary Data 7). The abundance of these modules was significantly decreased in children for whom POIT failed to induce either desensitization and/or remission (Figs. 2e and 4a). Notably, AA profiles were significantly different among POIT outcome groups at baseline ($n = 43$, $R^2 = 0.08$; $P = 0.006$, Fig. 4b) and at the end of avoidance ($n = 43$, $R^2 = 0.07$; $P = 0.039$, Fig. 4c), but not at the end of treatment (Two-sided PERMANOVA analyses, Supplementary Fig. 3f), suggesting that lower dietary AA intake and/or enhanced microbial AA metabolism during treatment differentiate those who do or do not develop POIT-associated remission.

The four POIT response-associated metabolite modules (UMM4, UMM5, UMM7, and UMM50) contained a total of 87 AAs and their derivatives, 68 of these belonged to UMM4 (Supplementary Data 7), which was significantly reduced in the D−R− group (Fig. 2e). The majority of these AA metabolites' abundances were decreased in relative concentration in the feces of children who failed to achieve remission (Fig. 4d). Notably, increased concentrations of microbial-derived branched-chain AA fermentation end products such as skatol and indole[33] were evident in both groups who failed to achieve remission (D+R− and D−R−) indicating that treatment failure may be due, in part, to increased microbial AA utilization.

AAs represent a major energy source for anaerobic gut bacteria[34] and select microbes are capable of harvesting AAs by deconjugating primary BAs[35]. Decreased fecal AA concentrations (Fig. 4d) and increased anaerobic energy and gluconeogenesis metabolism (Fig. 3b) in POIT-treated children who failed to achieve remission prompted us to investigate whether their fecal microbiomes encoded a distinct or enhanced capacity for AA or protein utilization. Using longitudinal microbial pathway abundance data, we found that pathways for L-histidine degradation, anaerobic energy metabolism, L-citrulline biosynthesis (Arginine degradation), and gluconeogenesis (which uses non-carbohydrate sources, including AA for energy production[36]) were enriched in the fecal microbiomes of children who did not achieve remission. In contrast, children who achieved remission possessed fecal microbiota enriched for pathways involved in AA biosynthesis (LME, $P < 0.05$, but *P.FDR* > 0.05; Supplementary Fig. 3g and Supplementary Data 14). Additionally, in our integrative MOFA2 analysis, Factor 3 significantly distinguished the D+R+ group from D+R− and D−R− groups (Supplementary Fig. 3b); L-tyrosine and L-tryptophan biosynthesis pathways were amongst the top five contributors to this factor's weight (Supplementary Fig. 3c). These data indicate that microbiomes with enhanced capacity for gluconeogenesis and AA metabolism associate with fecal AA metabolite depletion, which is characteristic of POIT-treated children who fail to achieve peanut allergy remission.

Differential abundance analysis of microbial enzymes at each time point revealed that only one differed between POIT responders and

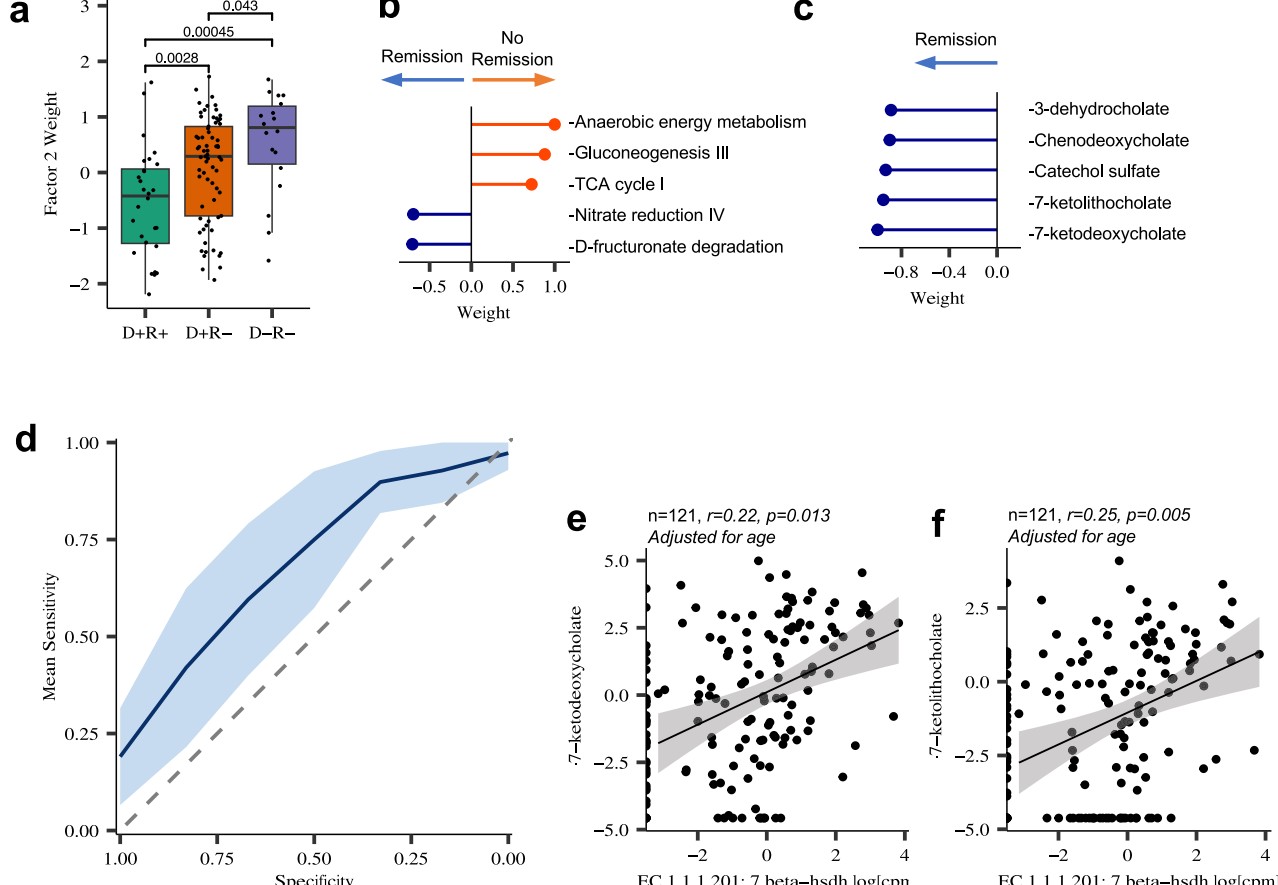

**Fig. 3 | Relative abundance of select pre-treatment bile acids predicts peanut oral immunotherapy outcomes. a** Factor 2 from MOFA2 analyses is the most significant differential factor between POIT response groups and weighted significantly higher in no remission groups compared to D+R+ group (D+R+ vs. D+R−, $P = 0.0028$; D+R+ vs. D−R−, $P = 0.00045$, two-sided Wilcoxon signed-rank test) (D+R+, $n = 28$, D+R−, $n = 69$, and D−R−, $n = 18$). Data are presented as mean values ± SEM. Boxplots show the median (center line), 25th and 75th percentiles (box bounds), and whiskers extend to values within 1.5× the interquartile range. **b** Top 5 microbial pathways contributing to the Factor 2 weight contain gluconeogenesis and anaerobic energy metabolism pathways. **c** Top 5 metabolites contributing to the Factor 2 weight are BA metabolites including 7-ketolithocholate and 7-ketodeoxycholate (D+R+, $n = 28$, D+R−, $n = 69$, and D−R−, $n = 18$). **d** The predictive ability of the logistic regression model is expressed as the Area Under Curve (AUC) which is computed from 100 times repeated five-fold cross-validation. Blue line shows the average across the 100 times repeated five-fold cross-validations with the shaded area representing the 95% CI (mean AUC ± standard deviation). The dashed diagonal line represents random chance. **e** The copy number of EC. 1.1.1.201 positively correlates with 7-ketodeoxycholate ($n = 121$, $r = 0.22$, $p = 0.013$. Adjusted for age), and **f** 7-ketolithocholate ($n = 121$, $r = 0.25$, $p = 0.005$, Adjusted for age). Two-sided Pearson correlation. The shaded area around the line represents the 95% confidence interval for the fitted regression line. Source data are provided as a Source Data file.

non-responders at the end of avoidance (LME, log2 fold change ≥ |1|, $P.FDR < 0.05$). No other microbial enzymes reached significance at other time points (Supplementary Data 15). The children who failed to develop POIT-induced remission had an increased copy number of the Xaa-Xaa-Proline tripeptidyl-peptidase gene (*ptpA*; Fig. 4e, *P.FDR <* 0.05), a hydrolase that cleaves N-terminal tripeptides with a proline residue at the third position[37]. The copy number of the *pptA* gene correlated with several modules including UMM7 AA (two-sided Pearson correlation, $R^2 = -0.26$; *P.FDR* = 0.058) and UMM10 Secondary BA (two-sided Pearson correlation, $R^2 = 0.28$; *P.FDR* = 0.058) modules (Supplementary Data 16).

Ara h 2, the most potent allergenic component of peanut protein[38], contains six proline residues[39]. In non-responsive children's fecal microbiomes, we found that the *ptpA* gene is encoded primarily by *Bacteroides* species including *B. dorei, B. uniformis, B. caccei* and *B. xylanisolvens* (Supplementary Fig. 3h). Therefore, we tested whether a correlation between bacterial *ptpA* copy number and Ara h 2-specific IgE levels existed in IMPACT participants who received POIT and identified a significant positive correlation at

baseline (Fig. 4f) end of the treatment (Fig. 4g), and at the end of avoidance (Fig. 4h). These data suggest that the fecal microbiomes of children who did fail to achieve POIT-induced remission have an increased capacity to metabolize proline-containing proteins, leading us to speculate that this may extend to the proteolysis-resistant Ara h 2 proteins necessary to promote remission of peanut allergy.

The learning early about peanut allergy (LEAP) clinical trial clearly demonstrated the critical importance of peanut exposure to develop immunological tolerance to peanut antigens[3,39]. Therefore, we hypothesized that the fecal microbiomes of children for whom POIT failed to induce remission have increased capacity for peanut protein degradation, effectively reducing antigen exposure. To test this, stabilized in vitro fecal microbiome cultures from participants in each of the outcome groups (D+R+, $n = 12$, D+R−, $n = 12$, D−R−, $n = 12$) were developed as previously described[40] and co-incubated with peanut extract under anaerobic conditions prior to Ara h 2 quantification by ELISA. Fecal microbiomes of all participants, regardless of remission outcome, exhibited the capacity to degrade Ara h 2, one of the most proteolytically resistant peanut protein antigens[41]. Moreover, peanuts

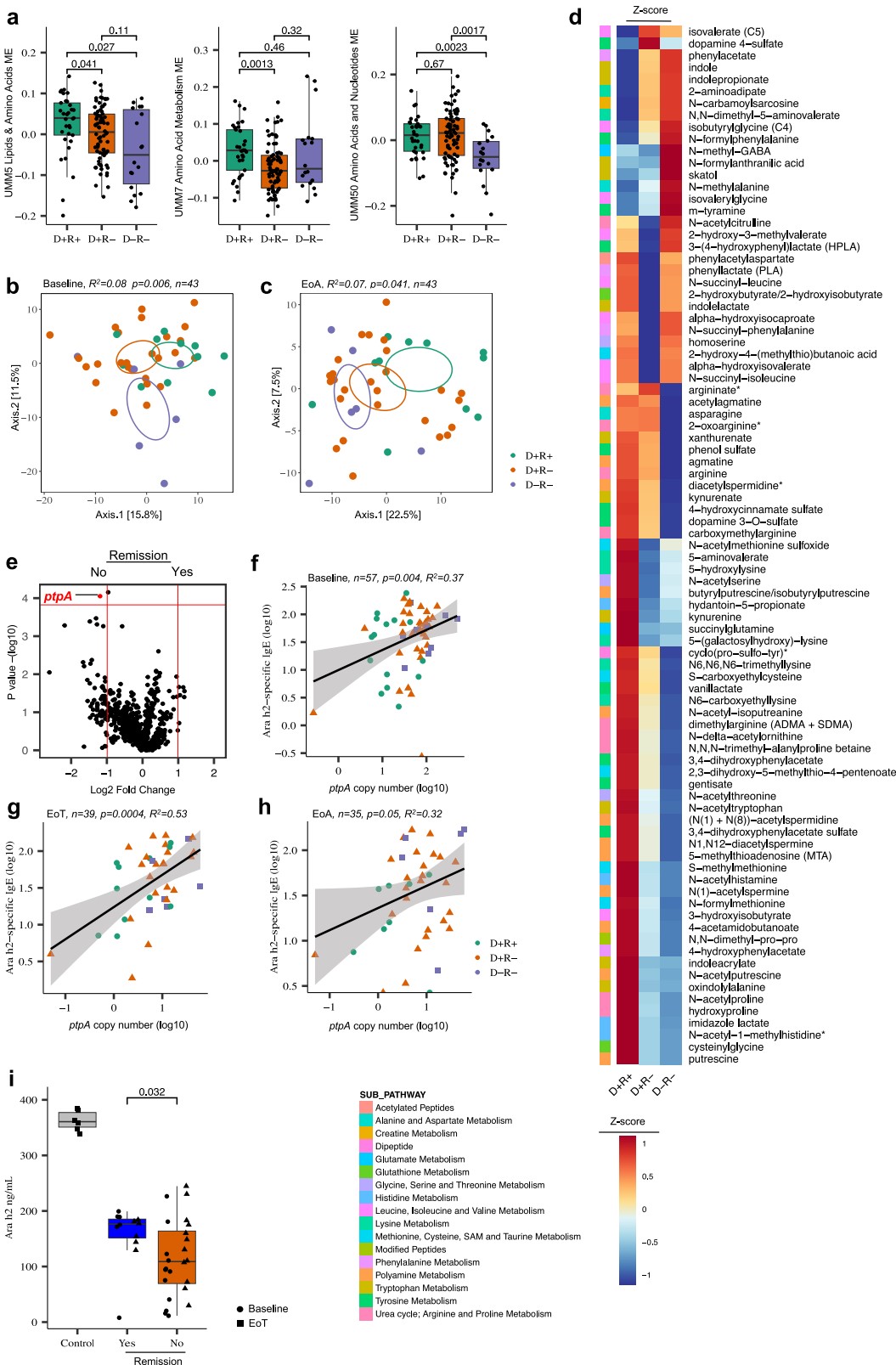

incubated with feces of patients who failed to achieve remission exhibited significantly decreased concentrations of this antigen compared to those who achieved remission. These data suggest that fecal conditions of those who fail to achieve remission following POIT, promote increased degradation of Ara h 2 proteins and thus endogenously reduce exposure to this key peanut antigen required for desensitization (Fig. 4i).

## Discussion

Microbiome analyses of fecal samples longitudinally collected from 90 IMPACT participants (Placebo, $n = 23$; POIT, $n = 57$) provide evidence that microbiome composition, function, and metabolic activities relate to POIT outcomes and that these relationships are both evident and strongest prior to treatment initiation. Although the fecal bacterial diversity and overall composition were similar between POIT and

**Fig. 4 | Enhanced microbiome protein metabolism associated with peanut oral immunotherapy failure. a** Difference in module eigenvectors of UMM5, UMM7 and UMM50 AA modules between POIT-outcome groups ($n = 129$; D+R+ = 33, D+R− = 78, D−R− = 18). Data are presented as mean values ± SEM. Boxplots show the median (center line), 25th and 75th percentiles (box bounds), and whiskers extend to values within 1.5× the interquartile range. Statistical comparisons were performed using the two-sided Wilcoxon rank-sum test. **b** Fecal AA metabolite composition is distinct between POIT outcome groups at **b**, baseline ($n = 43$, $R^2 = 0.08$; $P = 0.006$), and **c** end of avoidance ($n = 43$, $R^2 = 0.07$; $P = 0.041$). PERMANOVA analyses (two-sided) based on Euclidean distance matrix. **d** $Z$-scores of all AA metabolite abundance in nine modules associated with POIT outcomes. **e** POIT responders have an elevated *ptpA* gene copy number in their fecal microbiome at the end of avoidance compared to POIT non-responders ($P.FDR < 0.05$, two-sided linear mixed-effects models adjusted for multiple comparisons). Exact $P$ values are provided in Supplementary Data 15. **f** *ptpA* gene copy number positively correlates with Ara h2-specific IgE levels at baseline ($n = 57$, $p = 0.004$, $R^2 = 0.37$), **g** end of treatment ($n = 39$, $p = 0.0004$, $R^2 = 0.53$) and **h** end of avoidance ($n = 35$, $p = 0.05$, $R^2 = 0.32$). Two-sided Pearson correlation. **i** Fecal microbiome from children who did not achieve POIT-induced remission showed a greater capacity to metabolize peanut Ara h 2 protein than those who achieved remission ($P = 0.032$). Ara h 2 concentrations shown are averages of two independent experiments per sample. Each data point represents one biologically independent participant (remission $n = 6$, no remission = 12), and comparisons were made using a two-sided Wilcoxon rank-sum test. Boxplots show the median (center line), 25th and 75th percentiles (box bounds), and whiskers extend to values within 1.5× the interquartile range. Statistical comparisons were performed using the two-sided Wilcoxon rank-sum test. The control group refers to the BHI medium supplemented with peanut extract and incubated for 48 h with other samples without the microbiome inoculation. BA: bile acid. AA: amino acid. Source data are provided as a Source Data file.

placebo arms throughout the trial, within the POIT arm, fecal microbiota composition and functional capacity were distinct over a 3-year treatment period in those who do or do not achieve peanut allergy remission. BAs, including secondary BAs enriched in baseline samples associate with POIT-induced remission and appear to serve as a reasonable predictor of treatment outcome. Given the protracted nature of POIT and the high risk of severe adverse events in the treated peanut allergic population[4,5], utilization of fecal biomarkers to test for treatment responsiveness prior to initiation of therapy holds significant utility. Our findings are also consistent with a recent report showing that plasma BAs are associated with peanut oral immunotherapy efficacy in a smaller cohort of 20 children[40]. Gut microbial-derived secondary BAs act as hormones that regulate cholesterol metabolism, lipid and soluble vitamin uptake and influence energy balance via nuclear and G-protein-coupled receptors[41–43] that shape innate immune response[44,45]. Previous studies have demonstrated that the BA pool regulates colonic FOXP3+ regulatory T (Treg) cells that express the transcription factor RORγ[46]. Indeed, in a sub-analysis of the IMPACT trial participants ($n = 29$), Calise and colleagues examined T-cell profiles of peanut-challenged patients and found a trend towards increased expression of genes associated with regulatory T-cell function in desensitized patients compared to those who failed to achieve desensitization and remission[47].

Gut microbes are capable of metabolizing dietary proteins[48], utilizing host and dietary amino acids (AAs) for protein synthesis[49], and harvesting AAs from primary BAs to fuel central metabolism[50]. From our study, the gut microbiome of children for whom POIT failed to induce remission is characterized by enhanced microbial AA-utilization pathways, depletion of AAs and their metabolites, and enrichment of specific deconjugated secondary BA metabolites. Primary BAs, such as glycocholate and taurocholate, are typically conjugated to the AAs glycine and taurine[51], though a more recent study has indicated that a much broader range of AAs may be conjugated[52]. Primary BAs are converted to immunoregulatory secondary BAs by colonic bacteria[51]. Notably, recent studies have shown that primary BAs, particularly chenodeoxycholic acid, can promote food sensitization by activating retinoic acid response elements in dendritic cells, driving food allergen-specific IgE and IgG1 production[53]. Increased gut microbiome capacity to harvest AAs from conjugated BAs may elevate levels of deconjugated BAs, thus promoting allergic immune function and contributing to POIT failure.

Prior studies have established a strong association between impaired or delayed microbiota diversification over the first year of life and the onset of pediatric atopy[25,54,55]. These data indicate that a diverse gut microbiome during infancy is essential for appropriate immune development and the prevention of allergic disease. However, our findings revealed that amongst older peanut-allergic children, those who achieved POIT-induced remission exhibited significantly lower baseline microbiota diversity compared to those for whom POIT failed to induce remission. This observation appears to contrast with existing literature. However, we previously demonstrated that while infants at higher risk of allergic disease initially demonstrate lower fecal diversity compared to their lower-risk counterparts over the first year of life, between 18 and 24 months of age a cross-over event in fecal diversification occurs, with the former exhibiting sustained diversification while the latter reach an asymptote[56]. These data indicate that while lower fecal diversity in infancy is a consistent feature of those on the trajectory to allergy, in later childhood, higher fecal diversity is characteristic of those at increased risk of disease. Our study offers mechanisms as to why this may occur. Features associated with non-remission such as AA auxotrophy (i.e. inability to synthesize AAs required for growth), have previously been associated with higher microbiome diversity[57] due to microbial reliance on exogenous AA sources. In addition, we show that fecal BA profiles correlate with microbial diversity in peanut-allergic children, consistent with previous studies demonstrating that the BA pool and diversity regulate gut microbial composition and function[26]. Hence, while decreased fecal microbiota diversity in very early life represents a reproducible characteristic across all atopic pathologies, within older peanut-allergic children, BA and AA pools appear to promote greater fecal diversity and, more importantly, enhanced microbial capacity for AA and peanut protein metabolism associated with failure to achieve disease remission.

Previous studies have reported that members of the *Clostridia*, including *R. gnavus*, are associated with the development of food allergies[9,58–63]. The abundance of this species is positively correlated with age and is known to be capable of producing immunomodulatory secondary BAs[64]. In the IMPACT trial, participants who developed POIT-induced remission possessed a greater abundance of *R. gnavus*. These strains encoded enzymes that produce several of the specific BAs associated with disease remission suggesting that strain-specific BA metabolism capacity and not simply species relative abundance is paramount to food allergy clinical outcomes. Indeed, the metabolic capacity of *R. gnavus* strains is known to be large; previous studies indicate that strains from children with food allergies encode reduced fiber-degrading capacity and genes linked to pro-inflammatory polysaccharide production[9], distinguishing them from strains found in their non-allergic counterparts. These findings underscore the need for assessing strain-resolved functional differences that underlie food allergy phenotypes.

Food allergies and intolerances are typically triggered by specific protein motifs in foods such as Ara h proteins in peanuts, casein and beta-lactoglobulin in cow's milk, and tropomyosin proteins in shellfish[65]. Ingested allergens undergo enzymatic breakdown in the oral cavity, stomach, and small intestine[66] prior to interacting with antigen-presenting cells[67]. However, certain key antigenic peanut proteins, e.g. Ara h 2, are highly resistant to proteolysis[38], making it likely that they survive transit through the upper gastrointestinal tract

to the distal colon which houses the highest density of microbes and immune cells, including T and B effector cells[68]. The extent of peanut protein digestion determines the concentrations and profile of antigenic peanut peptides available for presentation by antigen-presenting cells, a key requirement for immune tolerance development. The distal gut is colonized by a complex community of metabolically active microbes capable of metabolizing dietary proteins[48,69,70], including food allergens. Our data indicates that increased fecal microbial peptidase activity, peanut protein degradation, and depletion of select immunomodulatory BA and AA during the peanut introduction period are associated with treatment failure. Previous studies demonstrated that microbe-free fecal extracts of infants who subsequently develop atopy or asthma promote canonical features of allergic inflammation in vitro, indicating that fecal metabolites are sufficient to drive allergic inflammation. Our more recent work showed that an atopy-predictive fecal microbial-derived lipid, 12,13 di-HOME exacerbates the inflammatory response to food allergens, including peanut, by promoting macrophage expression of *IL-1β, TNFα, NFkB,* and *IL-6*, expansion of memory B cells and increasing the ratio of IgE to IgG in peanut stimulated co-cultures[20]. Hence, the emerging data points to a more complex model in which the metabolic context in which antigenic stimulation occurs, governs functional immune response to allergen exposure and indicates that microbial processes impacting both antigen availability and immunomodulatory metabolites play a key role in dictating allergic outcomes.

Elucidating the impact of gut microorganisms on allergic food proteins may pave the way towards the development of more effective immunotherapeutic approaches by both targeting gut microbiome metabolic functions and protecting immunotherapeutic peanut proteins from microbial metabolism by, for example, encapsulating them in food-grade colloidal systems. A similar encapsulation system for gluten immunotherapy is currently being tested in several clinical trials[71,72], which so far have demonstrated safety and efficacy. Our study highlights the potential role of the gut microbiome in shaping POIT efficacy outcomes and suggests that specific fecal BAs could serve as both a prognostic biomarker to identify those for whom POIT may be most successful and as a therapeutic target to improve rates of POIT-induced remission.

## Methods

### Ethics statement
The IMPACT clinical trial was approved by the Office of Human Research Ethics (OHRE), University of North Carolina, Chapel Hill on April 9, 2013. The parent study titled, "IMPACT: Oral Immunotherapy (OIT) for Induction of Tolerance and Desensitization in Peanut-Allergic was a randomized, double-blind, placebo-controlled, multi-center study comparing peanut oral immunotherapy (OIT) to placebo. Informed consent was obtained from a parent or guardian of all participants. This study is an exploratory, non–pre-specified secondary analysis of the IMPACT clinical trial (NCT01867671), approved by the original study investigators.

### Clinical trial description and study population
Full details of the IMPACT clinical trial (NCT01867671) have been previously described[6].

### Sample collection and numbers
Stool samples were collected by participants at home and stored at clinical collection sites (Chapel Hill, NC; Little Rock, AR; Palo Alto, CA; Baltimore, MD; New York, NY) at −80 °C. Of the 146 participants enrolled in the IMPACT clinical trial (intention-to-treat group), 93 completed the treatment through the avoidance phase (per-protocol group). A total of 388 fecal samples were collected from 144 participants. Among the 93 participants who completed the treatment (per-protocol group), 327 fecal samples were obtained from 90 participants

−245 from the POIT group and 82 from the placebo group (57 and 23 participants, respectively, Supplementary Data 1). Three per-protocol participants did not provide fecal samples at any time point. One per-protocol participant who did not develop POIT-induced desensitization but developed remission (D−R+) was excluded (5 fecal samples from 5 time points) from all data analyses as a single sample was insufficient for statistical analyses. Frozen fecal samples were transferred on dry ice to the Lynch lab at the University of California, San Francisco, where all laboratory analyses were performed, except for untargeted metabolomics analyses, which were conducted at Metabolon Inc. in Morrisville, NC.

To maintain blinding, investigators did not have access to participant data until after 16S rRNA sequencing was completed and locked. As a result, all 388 fecal samples underwent 16S rRNA sequencing. High-quality 16S rRNA sequencing data were successfully generated for only 263 fecal samples from 79 participants, as some samples failed due to insufficient DNA, failed PCR, or did not pass the quality filtering and rarefaction (35,000 reads/samples). For analyses assessing the relationship between fecal microbiota composition and POIT outcomes, only samples from the per-protocol participants were included. However, baseline analyses, such as those presented in Fig. 1h–j, incorporated samples from all participants. In these cases, POIT outcomes were not a consideration, as the goal was to evaluate correlations between baseline bacterial phylogenetic diversity and serum IgE levels in peanut-allergic children.

For shotgun metagenomics, we focused on three key time points (baseline, end of treatment, and end of avoidance) while excluding mid-maintenance and end-of-build-up samples. This decision was based on cost considerations and the significant associations observed between fecal microbiota and different clinical outcomes at these three time points (Supplementary Data 4). DNA extracted for 16S rRNA sequencing with at least 100 ng of remaining material from these three time points was used for shotgun metagenome sequencing. Placebo participants who provided samples at baseline but did not provide fecal samples at the other two key time points (8 participants, Supplementary Fig. 1a) were excluded because we observed no significant differences in baseline fecal bacterial composition and diversity between the POIT and Placebo groups. High-quality shotgun metagenomics data were obtained from 80 participants (184 samples).

Finally, metabolomics data were generated for 58 participants who had corresponding shotgun metagenomics data and sufficient remaining material from all three key time points. Two participants were excluded because their samples were fully utilized during DNA extraction. Additionally, 20 participants who provided only baseline samples without subsequent time points were excluded to enable longitudinal metagenomics and metabolome integrative analyses with matching patient IDs. Thus, we retained data for 58 participants, 22 fewer than the number analyzed for shotgun metagenomics (Supplementary Fig. 1a and Supplementary Data 1).

### DNA extraction, 16S rRNA library preparation and sequencing
DNA was extracted from randomized fecal samples and positive controls (cat# D6300. ZymoBIOMICS Microbial Community Standard) using a modified cetyltrimethylammonium bromide (CTAB) buffer extraction protocol as previously described[73]. The variable region 4 (V4) of the 16S rRNA gene was amplified using 1 ng μl⁻¹ of DNA template using 515F and 806R primer pairs as previously described[74]. Amplicon concentrations were normalized using SequalPrep™ Normalization Plate Kit (Thermofisher Scientific), quantified using the Qubit 2.0 Fluorometer and the dsDNA HS Assay Kit (Life Technologies) and pooled at 5 ng per sample which was purified using AMPure SPRI beads (Beckman Coulter). 2 nM of the library was spiked with 30% of PhiX control v3 (Illumina). The denatured libraries and PhiX were diluted to 20 pM, and 1.5 pM were loaded onto the Illumina NextSeq 500/550 v2.5 High Output cartridge.

Sequence data was processed as previously described[75]. Forward and reverse reads were demultiplexed by using Quantitative Insights Into Microbial Ecology (QIIME 1.9.1)[76]. Sample sequences with more than two bases having a $Q$-score < 30 were truncated. As recommended by the Divisive Amplicon Denoising Algorithm 2 (DADA2) v1.16 protocol in R with the following modifications: Reads were maintained if they exhibited a maximum expected error of two and a read length of at least 150 base pair (bp) using the *filterAndTrim* function in the *dada2* package[77]. Reads were dereplicated and errors were learned on $1 \times 10^8$ reads, from samples chosen at random. Finally, chimeras were identified using the "consensus" method. Paired reads were merged with a minimum overlap of 25 bp, and reads were aggregated into a count table. Any V4 sequences abnormally short or long (±5 bp from the most frequently observed bp length; here: 253 bp) were also removed. We assigned taxonomic classifications to Sequence Variants (SVs) using *assignTaxonomy* in the *dada2* package and an 80% bootstrap cutoff, utilizing the SILVA v132 database[78], and species identification with *assignSpecies* at 100% identity. All species achieving an exact match were recorded, and the first in the list was used for descriptive purposes. Once these steps were completed for each run, all runs were combined into a complete SV table. A phylogenetic tree was constructed using *phangorn*[79] and *DECIPHER* packages[80]. The SV table was then filtered only to variants belonging to the kingdom Bacteria. Variants were also removed if they were present in less than 0.001% of the total number of observed sequence reads. Next, we employed methods to remove potential contaminants based on SVs present in negative controls. Specifically, SVs were removed if they were present in >15% of the negative controls and less than 15% of the samples[75] (primarily *Pseudomonas* SVs). For the remaining sequence variants in negative controls, the mean of the read count for each was calculated, rounded upward to the nearest whole number, and subtracted for each of these SVs in the dataset. Any remaining negative control SVs were subtracted from samples using the maximum read count across negative controls. Data was representatively rarefied at 35,000 reads per sample, a level selected to optimize sample count and community coverage.

### Metagenomic processing and data analysis

One-hundred eighty-four samples ($n = 75$ Baseline, $n = 54$ EoT, $n = 55$ EoA, Fig. 1a and Supplementary Data 1), were chosen among the DNA samples extracted for 16S rRNA sequencing including samples that went through untargeted metabolomic analyses. Fecal samples selected had sufficient remaining material for paired metagenomic and metabolomic profiling. DNA concentration was measured using the QuantiFluor dsDNA System on a Quantus Fluorometer (Promega, Madison, WI, USA). A Kapa Biosystems HyperPlus kit (Kapa Biosystems, Wilmington, MA, USA) was used for library construction. Briefly, 50 ng of genomic DNA was enzymatically sheared according to the manufacturer's instructions. DNA fragment ends were repaired, 3' adenylated, and ligated to adapters. The resulting adapter-ligated libraries were PCR-amplified. The PCR product was cleaned up from the reaction mix with magnetic beads. Then, Illumina libraries were quantified using the Qubit 2.0 Fluorometer with the dsDNA High Sensitivity Assay Kit (Life Technologies, Grand Island, NY) and pooled at equal molar concentrations. The final pooled libraries were submitted to the Center for Advanced Technology (CAT) at the University of California San Francisco. The pooled libraries were sequenced using the Illumina NovaSeq 6000 in a 2 × 150 bp paired-end run protocol targeting a minimum of 60,000,000 raw reads per sample in total.

Raw sequences from all lanes were merged into a concatenated file for each sample. Raw FASTQ files underwent FASTQC[81] and quality and contaminant filtering using *bbTools* v38.73. Specifically, *bbduk* (v38.73) (https://sourceforge.net/projects/bbmap/) trimmed Illumina adapters, removed any PhiX contamination, filtered low-quality sequences, and employed trimming after a $Q$ score <15 from both

the 3' and 5' directions. Finally, *bbmap* removed reads mapping to the human genome using GRCh38[82] as the reference database. The median number of raw reads per sample was 97,502,238 (IQR 30,132,152). All analyses were performed on quality-filtered reads. HUMAnN 3.0 pipeline was used to identify genes[83], level4ECs, and functional Meta-Cyc pathways from the short-reads, and to normalize outputs into copies per million (CPM). MetaCyc reactions and level4ECs enzymes that were present in <20% of samples were removed and, yield 517 MetaCyc reactions and 2605 level4ECs enzymes for downstream analyses.

Taxonomic classification of metagenomic reads was conducted using Kraken2, a k-mer-based tool designed to efficiently and accurately assign reads to taxonomic labels. The Kraken2 database (v.2021), encompassing genomes from bacteria, archaea, viruses, fungi, and other eukaryotes, was used for classification. The analysis was conducted on paired-end reads, using a confidence threshold of 0.95 to ensure robust taxonomic assignments. The raw taxonomic counts for each sample were imported into a phyloseq object for downstream analysis.

### Untargeted metabolomics analyses

Among the samples that went through shotgun-metagenome analyses, 174 ($n = 58$ Baseline, $n = 58$ EoT, $n = 58$ EoA, Fig. 1a, and Supplementary Data 1) matching samples were available for untargeted metabolomics analyses. Two hundred milligrams of stool per sample was submitted to Metabolon Inc. (Durham, NC) for ultrahigh performance liquid chromatography/tandem mass spectrometry (UPLC–MS/MS) and gas chromatography–mass spectrometry (GC–MS) using their standard protocol (http://www.metabolon.com/).

### Sample accessioning

Following receipt, samples were inventoried and immediately stored at −80 °C. Each sample received was accessioned into the Metabolon LIMS system and was assigned by the LIMS a unique identifier that was associated with the original source identifier only. This identifier was used to track all sample handling, tasks, results, etc. The samples (and all derived aliquots) were tracked by the LIMS system. All portions of any sample were automatically assigned their own unique identifiers by the LIMS when a new task was created; the relationship of these samples was also tracked. All samples were maintained at −80 °C until processed.

### Sample preparation

Samples were prepared using the automated MicroLab STAR® system from Hamilton Company. Several recovery standards were added prior to the first step in the extraction process for QC purposes. To remove protein, dissociate small molecules bound to protein or trapped in the precipitated protein matrix, and to recover chemically diverse metabolites, proteins were precipitated with methanol under vigorous shaking for 2 min (Glen Mills GenoGrinder 2000) followed by centrifugation. The resulting extract was divided into five fractions: two for analysis by two separate reverse phase (RP)/UPLC–MS/MS methods with positive ion mode electrospray ionization (ESI), one for analysis by RP/UPLC–MS/MS with negative ion mode ESI, one for analysis by HILIC/UPLC–MS/MS with negative ion mode ESI, and one sample was reserved for backup. Samples were placed briefly on a TurboVap® (Zymark) to remove the organic solvent. The sample extracts were stored overnight under nitrogen before preparation for analysis.

### QA/QC

Several types of controls were analyzed in concert with the experimental samples: a pooled matrix sample generated by taking a small volume of each experimental sample (or alternatively, use of a pool of well-characterized human plasma) served as a technical replicate throughout the data set; extracted water samples served as process

blanks; and a cocktail of QC standards that were carefully chosen not to interfere with the measurement of endogenous compounds were spiked into every analyzed sample, allowed instrument performance monitoring and aided chromatographic alignment. Instrument variability was determined by calculating the median relative standard deviation (RSD) for the standards that were added to each sample prior to injection into the mass spectrometers. Overall process variability was determined by calculating the median RSD for all endogenous metabolites (i.e., non-instrument standards) present in 100% of the pooled matrix samples. Experimental samples were randomized across the platform run with QC samples spaced evenly among the injections.

## Ultrahigh performance liquid chromatography–tandem mass spectroscopy (UPLC–MS/MS)

All methods utilized a Waters ACQUITY ultra-performance liquid chromatography (UPLC) and a Thermo Scientific Q-Exactive high resolution/accurate mass spectrometer interfaced with a heated electrospray ionization (HESI-II) source and Orbitrap mass analyzer operated at 35,000 mass resolution. The sample extract was dried and then reconstituted in solvents compatible with each of the four methods. Each reconstitution solvent contained a series of standards at fixed concentrations to ensure injection and chromatographic consistency. One aliquot was analyzed using acidic positive ion conditions, chromatographically optimized for more hydrophilic compounds. In this method, the extract was gradient eluted from a C18 column (Waters UPLC BEH C18-$2.1 \times 100$ mm, $1.7 \, \mu$m) using water and methanol, containing 0.05% perfluoropentanoic acid (PFPA) and 0.1% formic acid (FA). Another aliquot was also analyzed using acidic positive ion conditions, however, it was chromatographically optimized for more hydrophobic compounds. In this method, the extract was gradient eluted from the same aforementioned C18 column using methanol, acetonitrile, water, 0.05% PFPA, and 0.01% FA and was operated at an overall higher organic content. Another aliquot was analyzed using basic negative ion optimized conditions using a separate dedicated C18 column. The basic extracts were gradient eluted from the column using methanol and water, however with 6.5 mM Ammonium Bicarbonate at pH 8. The fourth aliquot was analyzed via negative ionization following elution from a HILIC column (Waters UPLC BEH Amide $2.1 \times 150$ mm, $1.7 \, \mu$m) using a gradient consisting of water and acetonitrile with 10 mM Ammonium Formate, pH 10.8. The MS analysis alternated between MS and data-dependent $MS^n$ scans using dynamic exclusion. The scan range varied slightly between methods but covered 70–1000 $m/z$. Raw data files are archived and extracted as described below.

## Bioinformatics

The informatics system consisted of four major components, the Laboratory Information Management System (LIMS), the data extraction and peak-identification software, data processing tools for QC and compound identification, and a collection of information interpretation and visualization tools for use by data analysts. The hardware and software foundations for these informatics components were the LAN backbone, and a database server running Oracle 10.2.0.1 Enterprise Edition.

## LIMS

The purpose of the Metabolon LIMS system was to enable fully auditable laboratory automation through a secure, easy-to-use, and highly specialized system. The scope of the Metabolon LIMS system encompasses sample accessioning, sample preparation, instrumental analysis and reporting, and advanced data analysis. All of the subsequent software systems are grounded in the LIMS data structures. It has been modified to leverage and interface with the in-house information extraction and data visualization systems, as well as third-party instrumentation and data analysis software.

## Data extraction and compound identification

Raw data was extracted, peak-identified and QC processed using Metabolon's hardware and software. These systems are built on a web-service platform utilizing Microsoft's.NET technologies, which run on high-performance application servers and fiber-channel storage arrays in clusters to provide active failover and load-balancing. Compounds were identified by comparison to library entries of purified standards or recurrent unknown entities. Metabolon maintains a library based on authenticated standards that contain the retention time/index (RI), mass-to-charge ratio ($m/z$), and chromatographic data (including MS/MS spectral data) on all molecules present in the library. Furthermore, biochemical identifications are based on three criteria: retention index within a narrow RI window of the proposed identification, accurate mass match to the library $\pm 10$ ppm, and the MS/MS forward and reverse scores between the experimental data and authentic standards. The MS/MS scores are based on a comparison of the ions present in the experimental spectrum to the ions present in the library spectrum. While there may be similarities between these molecules based on one of these factors, the use of all three data points can be utilized to distinguish and differentiate biochemicals. More than 3300 commercially available purified standard compounds have been acquired and registered into LIMS for analysis on all platforms for determination of their analytical characteristics. Additional mass spectral entries have been created for structurally unnamed biochemicals, which have been identified by virtue of their recurrent nature (both chromatographic and mass spectral). These compounds have the potential to be identified by future acquisition of a matching purified standard or by classical structural analysis.

## Curation

A variety of curation procedures were carried out to ensure that a high-quality data set was made available for statistical analysis and data interpretation. The QC and curation processes were designed to ensure accurate and consistent identification of true chemical entities and to remove those representing system artifacts, misassignments, and background noise. Metabolon data analysts use proprietary visualization and interpretation software to confirm the consistency of peak identification among the various samples. Library matches for each compound were checked for each sample and corrected if necessary.

## Metabolite quantification and data normalization

Peaks were quantified using area-under-the-curve. For studies spanning multiple days, a data normalization step was performed to correct variation resulting from instrument inter-day tuning differences. Essentially, each compound was corrected in run-day blocks by registering the medians to equal one (1.00) and normalizing each data point. For studies that did not require more than one day of analysis, no normalization is necessary, other than for purposes of data visualization. In certain instances, biochemical data may have been normalized to an additional factor (e.g., cell counts, total protein as determined by Bradford assay, osmolality, etc.) to account for differences in metabolite levels due to differences in the amount of material present in each sample. For network and statistical analyses, normalized, imputed, and log-transformed areas under the curve dataset were used.

## In vitro fecal microbiome metabolism of peanut

Stool samples from IMPACT participants were prepared for culture as described previously[84]. Briefly, stool samples from 18 patients (D+R+, $n = 6$, D+R−, $n = 6$, D−R−, $n = 6$) at two time points (baseline and end of treatment) with sufficient paired baseline and end-of-treatment material for the experiment were thawed on ice. All fecal processing was completed under aerobic conditions. Stools were resuspended in Brain Heart Infusion (BHI) media at a ratio of 10 ml/g stool prior to

vigorous vertexing for 5 min and filtering with a 50 μm cell strainer and storage at −80 °C following 25% (volume/volume) glycerol addition. A total of 10 μL of prepared feces was used to inoculate 1 mL of BHI medium supplemented with 8 μL peanut extract (1/10 weight/volume in 50% glycerin, Hollister-Stier) and incubated for 48 h at 37 °C under anaerobic conditions. Following 48 h of incubation, microbiome cultures were centrifuged at 3200×g for 10 min and filtered through 0.22 μm filters. Ara h 2 peptide concentrations were determined using an Enzyme-Linked Immunosorbent Assay (ELISA) according to manufacturer instructions (Indoor Biotechnologies, Charlottesville, VA). Each fecal microbiome culture originated from a unique biologically independent participant ($n = 18$; Remission = 6, No Remission = 12), and two independent in vitro experiments were performed per sample. Statistical comparisons were performed on the mean Ara h 2 values per participant, thus reflecting inter-individual biological variability.

## Statistical analyses

Statistical analyses were performed in the R statistical programming language version 4.3.2. a-diversity indices with Faith's phylogenetic diversity, Pielou's evenness, and Chao1 species richness were calculated in QIIME and using the *vegan* v2.6-4 and *picante* v1.8.2 packages in R. For correlation analyses between baseline phylogenetic diversity and serum IgE levels, participants enrolled in the IMPACT trial but who did not complete the trial were included as 16S rRNA sequencing and clinical serum IgE levels were available. Wilcoxon rank-sum and signed-rank tests were performed in R using the wilcox.test function; by default, tests were two-sided unless otherwise specified in figure legends. For beta-diversity (microbiome composition), distance matrices based on unweighted UniFrac, weighted UniFrac, Bray–Curtis, and Canberra for 16S rDNA data and Euclidean for metabolomics dataset were generated using the *distance* function from *phyloseq* v1.30.0[85] and ordinated into two-dimensional space using the *pcoa* function from the *ape* v5.3 package[86]. Permutational analysis of variance tests (Two-sided PERMANOVA; $R^2$ and P values) were generated for independent terms with 1000 permutations using *adonis2* from the *vegan* package v2.6-4[87]. correlations and P values were calculated and corrected for potential confounding factors such as age at screening, using the cor.test function from the stats package v.4.3.2 in R. When samples were used from multiple time points, for example, in linear mixed-effect model (LME) models on longitudinal samples, only age was adjusted and stated in figure legends. LME models were conducted as two-sided tests unless explicitly stated otherwise.

## Generalized linear mixed-effect model

Generalized linear mixed-effect models were employed on longitudinal microbiome data to determine differences in microbial taxa, microbial pathways, metabolites between POIT outcome groups (D+R+, D+R−, D−R−) and remission outcome (yes or no), using a custom script (https://github.com/lynchlab-ucsf/lab-code/blob/master/SigTaxa/ManyModelScript.R) that employs multiple statistical models (Linear Model, Compound Poisson Linear Model, Poisson, Negative Binomial, and Tweedie) and compared using the AIC before reporting the final estimate and p-value. False-discovery corrections were made using the Benjamini-Hochberg method.

## Weighted gene correlation network analyses

Co-occurrence networks of microbial pathways and metabolites were constructed using weighted correlation network analysis (WGCNA) with the R package *WGCNA* v1.72-5[88] to find modules of highly interconnected, mutually exclusive metabolites. Pearson correlations were used to determine inter-metabolite and inter-microbial pathway relationships, where modules are composed of positively correlated metabolites. We constructed a signed network using specific parameters (power = 7, reassignThreshold = 0, mergeCutHeight = 0.25), by applying

hierarchical clustering and topology overlap measures (TOM). The minimum module size was set to five metabolites. Module eigenvectors (MEs) were defined as the first principal component of a given module and considered as a representative measure of the joint abundance profile of that module. Each module eigenvectors was used to test the association between its respective module and POIT-outcomes using ANOVA in all samples which are adjusted for participants' age. *P.FDR* < 0.05 is considered significant. Differences in MEs between POIT-outcome groups were tested and plotted using the Wilcoxon signed-rank test. Module membership was used to determine the inter-connectedness of each metabolite to its assigned module to identify "hub" metabolites: this was defined as the correlation between each metabolite and the module eigenvectors (MEs) (strong positive values indicate high interconnectedness) as previously described[11]. Metabolite heatmaps visualized in Figs. 2f–h, and 4d were generated using the *pheatmap* R package. Briefly, BA and AA metabolites from the POIT response-associated metabolomics modules (Fig. 2a) were filtered. Mean metabolite values were calculated across treatment groups (D+R+, D+R −, D−R−). To normalize metabolite abundance, Z-score scaling was applied across each group.

## Multi-omics factor analyses (MOFA2)

MOFA (v1.12.1) uses multi-omics data from the same set of samples as input and generates a model that infers a set of "Factors" that best explain patterns of covariation across samples[89]. Details of the methodology can be found in the original publication[90]. As input for the MOFA model, we used untargeted metabolomics (1538 metabolites) and shotgun metagenomics datasets (518 features). All inputs were normalized by centralized log normalization. When fitting the model, we selected the top factors ordered by the mean fractional variance explained across omic modalities (that is, factor 1 contributed the most, and factor 7 contributed the least to mean fractional variation; Supplementary Fig. 3a). When testing factor values for statistically significant differences between POIT outcome groups we used a two-tailed Mann–Whitney *U*-test. Supplementary Fig. 3b). Top five features of metagenome and metabolome datasets from significant factors were displayed (Supplementary Fig. 3c).

## Machine learning model with logistic regression and random forest

For the predictive metabolite analysis, normalized abundance of the top five metabolites from Factor 2 of the MOFA2 analyses was processed with the mikropml R package (v1.6.1) (https://CRAN.R-project.org/package=mikropml)[91]. We used Random Forest (rf) and Logistic Regression functions (glmnet) with Remission (Yes versus No) as an outcome using 50% of the samples as the training set and 50% as the test set. Model performances were evaluated with repeated *k*-fold cross-validation (tenfold, 10 repetitions) and parameters were tuned by choosing mtry and values between 1 and the square root of the total number of variables. Model training was accomplished with the caret R package v6.0-94 (https://topepo.github.io/caret/), mtry and lambda values that determined the highest model accuracy were chosen as input to random forest and logistic regression analysis, respectively. Variable importance was assessed with permutations (100 iterations). Full results are reported in Supplementary Data 13.

## Reporting summary

Further information on research design is available in the Nature Portfolio Reporting Summary linked to this article.

## Data availability

All data supporting the findings described in this manuscript are available in the article and in the Supplementary Information, Supplementary data files and Source data file. The 16S rRNA sequencing and shotgun metagenomic sequencing data generated in this study

have been deposited in the NCBI SRA database under accession codes PRJNA1261081 and PRJNA1261967, respectively. The untargeted metabolomics data is available at the NIH Common Fund's National Metabolomics Data Repository (NMDR) website, the Metabolomics Workbench[92], where it has been assigned Study ID (ST003917) and Project ID (PR002450)[93]. The data can be accessed directly via its Project https://doi.org/10.21228/M8Z84V. Source data are provided in this paper.

## Code availability

R scripts including Many Model (Generalized linear mixed-effect model) and PERMANOVA scripts used for data analyses and described in the "Methods" section were previously reported[94,95] and available at https://github.com/lynchlab-ucsf. All R codes used to generate figures in this study will be made available from the corresponding author upon request.

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

## Acknowledgements
Research reported in this publication was supported by the National Institute of Allergy and Infectious Diseases of the National Institutes of Health under Award Number UM1AI109565 and by S.V.L.'s research program which is funded by AI128482, AI148104, UM1AI160040 and AI089473. The content is solely the responsibility of the authors and does not necessarily represent the official views of the National Institutes of Health. M.Ö. is supported by postdoctoral T32 fellowship 2T32DK007762-46. We thank Slavena Vylkova, Rebecca L. Knoll, and Elad Deiss-Yehiely for their internal review of this article. We also thank the Immune Tolerance Network (ITN) Leadership, IMPACT study team, and trial participants for making this study possible.

## Author contributions
Conceptualization: M.Ö. and S.V.L.; methodology: M.Ö., D.L.L., C.L.G., A.L., J.C.G., and S.V.L.; data analysis: M.Ö.; investigation: M.Ö. D.L.L., C.L.G., A.L., L.M.W., C.B., S.S., S.M.J., and S.V.L.; writing and editing original draft: M.Ö. and S.V.L.; writing review: M.Ö., C.L.G., and S.V.L.; data visualization: M.Ö.; supervision: S.V.L.; funding acquisition: S.V.L.

## Competing interests
S.V.L. is a board member and consultant for the biotechnology company Siolta Therapeutics, Inc., and holds stock in the company. She also consults for Sanofi and for the Atria Institute of New York. S.S. has served on a scientific advisory board for Sanofi. S.M.J, reports grant support from the NIH-NIAID and Food Allergy Research & Education (FARE), clinical trials support Genentech, ALK-Abello, Inc., Aravax Pty, Ltd., DBV Technologies, Inc., Genentech, Inc., Novartis, Inc., Regeneron Pharmaceuticals, Inc., and personal fees from LAmAb Bio, Inc. M.Ö., D.L.L., C.L.D., A.L., J.C.G., L.M.W., and C.H.B. declare that they have no relevant competing interests.
