## [Transparent Peer Review file · Nature Communications]

Gut Microbial Bile and Amino Acid Metabolism Associate with Peanut Oral Immunotherapy Failure

Corresponding Author: Dr Susan Lynch

Version 0:

Reviewer comments:

Reviewer #1

(Remarks to the Author)

The authors have carefully, thoughtfully and thoroughly responded to all of the concerns raised in the initial review.

One additional concern - please check:

original manuscript line 373

"Our findings indicate that a gut microbiome that primarily derives energy from amino acid fermentation results in the depletion of immunomodulatory amino acid and secondary bile acids both of which associate with POIT remission failure."

revised manuscript line 452

"Our data indicates that increased fecal microbial peptidase activity, peanut protein degradation and depletion of immunomodulatory BA and AA during the peanut introduction period associate with treatment efficacy."

Is this an issue of timing? How can depletion of immunomodulatory SBA associate with both POIT remission failure and efficacy?

Reviewer #2

(Remarks to the Author)

I responded based on my response to reviewer 3's comments.

The authors have responded to the reviewers' comments as required, and most of the data presented in the revised manuscript is convincing.

1. At the end of the sentence line 46-48, the following text should be added ", with the exception of a few limited facilities conducting OIT as clinical research."
2. For Figure 1 and Figure 2, some of figures do not state at which period the specimens are compared. For example, H-I in Figure 1, C-E and I-L in Figure 2
3. Can the authors comment on which metabolite module the microbe with Xaa-Xaa-Prokine tripeptidyl-peptidase gene may be associated with?

Reviewer #3

(Remarks to the Author)

The data presented are well described and highly impactful to field. The demonstration of clear differences in the subject microbiota and metabolites prior to the start of OIT and their association to remission status generates a clear translational pathway to the investigation of adjunctive therapies that may reduce side effects associated with therapy and improve outcomes. I have no further comments for this manuscript but would welcome an analysis of side effects experienced by subject with the same metabolic differences.

Point-by-point responses to reviewers

Dear Reviewers

We sincerely thank you for your thoughtful and constructive feedback on our manuscript entitled "*Gut Microbial Bile and Amino Acid Metabolism Associate with Peanut Oral Immunotherapy Failure.*" We are grateful for the opportunity to revise our work and are pleased that the manuscript is considered favorably for publication.

We have carefully considered the reviewers' comments and have revised the manuscript accordingly. Below, we provide a detailed, point-by-point response to each comment. Reviewer comments are reproduced in italics, followed by our responses in regular text. Where appropriate, we have indicated changes made in the manuscript and referenced specific line numbers in the revised version.

REVIEWERS' COMMENTS

Reviewer #1 (Remarks to the Author): *The authors have carefully, thoughtfully and thoroughly responded to all of the concerns raised in the initial review. One additional concern - please check: original manuscript line 373 "Our findings indicate that a gut microbiome that primarily derives energy from amino acid fermentation results in the depletion of immunomodulatory amino acid and secondary bile acids both of which associate with POIT remission failure." revised manuscript line 452 "Our data indicates that increased fecal microbial peptidase activity, peanut protein degradation and depletion of immunomodulatory BA and AA during the peanut introduction period associate with treatment efficacy." Is this an issue of timing? How can depletion of immunomodulatory SBA associate with both POIT remission failure and efficacy?*

Thank you for highlighting this point. We have now changed this language in this sentence to clarify: "*Our data indicates that increased fecal microbial peptidase activity, peanut protein degradation and depletion of select immunomodulatory BA and AA during the peanut introduction period associate with treatment failure.*" Line 463-466

Reviewer #2 (Remarks to the Author): I responded based on my response to reviewer 3's comments. The authors have responded to the reviewers' comments as required, and most of the data presented in the revised manuscript is convincing.

1. At the end of the sentence line 46-48, the following text should be added “, with the exception of a few limited facilities conducting OIT as clinical research.”

That sentence has been revised accordingly.

2. For Figure 1 and Figure 2, some of figures do not state at which period the specimens are compared. For example, H-I in Figure 1, C-E and I-L in Figure 2

Time periods are now clarified on those figures or their legends. Thank you for pointing this out.

3. Can the authors comment on which metabolite module the microbe with Xaa-Xaa-Prokine tripeptidyl-peptidase gene may be associated with?

We have added an additional supplementary table showing two-sided Pearson correlation results between the copy number of the *ptpA* gene and the untargeted metabolomics modules listed below (two-sided Pearson correlation, *P.FDR* < 0.1 considered significant). (Supplementary table 16 and lines 348-349).

Module Name	R squared	P value	P.FDR
UMM7 Amino Acid Metabolism (Glu, Leu, Ile, Va)	-0.26	0.01	0.058
UMM10 Secondary Bile Acid Metabolism	0.28	0.00	0.058
UMM26 Amino Acids	-0.26	0.01	0.058
UMM35 Nucleotides, Amino Acids & Carbohydrates	-0.29	0.00	0.058
UMM41 Galactosyl Glycerolipids	-0.26	0.01	0.058
UMM25 Xenobiotics - Food Component/Plant	-0.25	0.01	0.063

Reviewer #3 (Remarks to the Author): The data presented are well described and highly impactful to field. The demonstration of clear differences in the subject microbiota and metabolites prior to the start of OIT and their association to remission status generates a clear translational pathway to the investigation of adjunctive therapies that may reduce side effects associated with therapy and improve outcomes. I have no further comments for this manuscript but would welcome an analysis of side effects experienced by subject with the same metabolic differences.

Thank you for the encouraging feedback. We agree that the analysis of side effects is an important and interesting direction. However, this manuscript focuses specifically on the relationship between gut microbial functional capacity and POIT efficacy and thus does not include an analysis of treatment-related side effects. We appreciate the suggestion and consider it a valuable avenue for future investigation.